# One-form symmetries in $\mathcal{N}=3$ $S$-folds

**Antonio Amariti[1], Davide Morgante[1,2], Antoine Pasternak[1],**
**Simone Rota[1,2] and Valdo Tatitscheff[3★]**

**1** INFN, Sezione di Milano, Via Celoria 16, I-20133 Milano, Italy
**2** Dipartimento di Fisica, Università degli Studi di Milano,
Via Celoria 16, I-20133 Milano, Italy
**3** Institut für Mathematik, Universität Heidelberg,
Im Neuenheimer Feld 205, 69120 Heidelberg, Germany

★ vtatitscheff@mathi.uni-heidelberg.de

## Abstract

We classify the global one-form symmetries for non-Lagrangian $\mathcal{N}=3$ SCFTs that arise by the action of $S$-fold projections on D3-branes. Such a classification is dictated, on a generic point of the Coulomb branch, by probing the charge spectrum of $(p,q)$-strings in the brane setup. The charge lattice of lines is then obtained by finding the ones that are genuine modulo screening by dynamical particles. The one-form symmetries are then extracted from the maximal sub-lattices of mutually local lines. We further comment on the existence of non-invertible symmetries for some of these $\mathcal{N}=3$ SCFTs.



# 1 Introduction

In the recent past, a new paradigm for the notion of symmetry in QFTs became dominant. It is based on the necessity to include higher-form symmetries and the corresponding extended objects in the description of quantum field theories [1]. Restricting to four-dimensional QFTs, the simplest way to proceed consists in classifying the one-form symmetries in supersymmetric and conformal theories (SCFTs). A seminal paper that allowed for such a classification has been [2] where a general prescription was given in terms of the spectrum of mutually local Wilson and 't Hooft lines [3]. Such a prescription was initially based on the existence of a Lagrangian description for the SCFT under investigation. In absence of a Lagrangian description it is nevertheless possible to use other tools, coming from supersymmetry, holography and/or branes. These constructions have allowed to figure out the one-form symmetry structure of many different QFTs, including some 4d non-Lagrangian SCFTs, see [4–20].

A class of theories that has not been deeply investigated so far are SCFTs with 24 supercharges, i.e. $\mathcal{N} = 3$ conformal theories. Such models have been predicted in [21], and then found in [22]. Many generalizations have been then studied by using various approaches [23–30]. A key role in the analysis of [22] is based on the existence, in the string theory setup, of non-perturbative extended objects that generalizes the notion of orientifolds, the $S$-folds (see [31, 32] for their original definition). From the field theory side, the projection implied by such $S$-folds on $\mathcal{N} = 4$ SYM has been associated to the combined action of an R-symmetry and an S-duality twist on the model at a fixed value of the holomorphic gauge coupling, where the global symmetry is enhanced by opportune discrete factors. Four possible $\mathbb{Z}_k$ have been identified, corresponding to $k = 2$, 3, 4 and 6. While the $\mathbb{Z}_2$ case corresponds to the original case of the orientifolds [33–38], where actually the holomorphic gauge coupling does not require to be fixed, the other values of $k$ correspond to new projections that can break supersymmetry down to $\mathcal{N} = 3$. The analysis has been further refined in [23], where the discrete torsion, in analogy with the case of orientifolds, has been added to this description. In this way, it has been possible to achieve a classification of such $\mathcal{N} = 3$ $S$-folds SCFT in terms of the Shephard–Todd complex reflection groups.

The goal of this paper consists in classifying one-form symmetries for such theories, constructing the lattices of lines and identifying which models possess non-invertible symmetries. The main motivation behind this expectation is that for the rank-2 $S$-folds, in absence of discrete torsion, the SCFTs enhance to $\mathcal{N} = 4$ SYM [23] where these properties are present. The existence of non-trivial one-form symmetries in some exceptional $\mathcal{N} = 3$ theories has also been argued in [39].

Our strategy adapts the one presented in [9] to $S$-fold setups. There, the spectrum of lines is built from the knowledge of the electromagnetic charges of massive states in a generic point of the Coulomb branch. These charges are read from the BPS quiver, under the assumption that the BPS spectrum is a good representative of the whole spectrum of electromagnetic charges. In the case of $S$-folds however such a BPS quiver description has not been worked out and we extract the electromagnetic charges of dynamical particles from the knowledge of the $(p, q)$-

Table 1: Summary of our results. $\mathbb{1}$ represents a trivial group.

| $S$-fold | One-form symmetry | # of inequivalent lattices | Non-invertible symmetry |
|---|---|---|---|
| $S_{3,1}$ | $\mathbb{Z}_3$ | 2 | Yes |
| $S_{3,3}$ | $\mathbb{1}$ | 1 | No |
| $S_{4,1}$ | $\mathbb{Z}_2$ | 2 | Yes |
| $S_{4,4}$ | $\mathbb{1}$ | 1 | No |
| $S_{6,1}$ | $\mathbb{1}$ | 1 | No |

strings configurations in the Type IIB setup [40,41]. The main assumption behind the analysis is that such charges are a good representative of the electromagnetic spectrum.

We proceed as follows. First we choose an $\mathcal{N} = 3$ theory constructed via an $S$-fold projection of Type IIB. This consists in having $N$ D3-branes, together with their images, on the background of an $S$-fold. At a generic point of the Coulomb branch, the corresponding low energy gauge dynamics corresponds to a $U(1)^N$ gauge theory where each $U(1)$ is associated to a D3. Then we list all $(p, q)$-strings that can be stretched between D3-branes and their images. They have electric and magnetic charges with respect to $U(1)^N$. Eventually we run the procedure of [9]. This consist in finding all the lines that are genuine, i.e. have integer Dirac pairing with the local particles, modulo screening by the dynamical particles. This gives the lattice of possible charges, then the different global structures correspond to maximal sub–lattices of mutually local lines.

Our results are summarized in Table 1. In the first column, one finds the type of $S$-fold projection that has been considered. Such projections are identified by the two integers $k$ and $\ell$ in $S_{k,\ell}$. The integer $k$ corresponds to the $\mathbb{Z}_k$ projection while the second integer $\ell$ is associated to the discrete torsion. Then, when considering an $S_{k,\ell}$ $S$-fold on a stack of $N$ D3-branes the complex reflection group associated to such a projection is $G(k, k/\ell, N)$. In the second column, we provide the one-form symmetry that we found in our analysis, and in the third, the number of inequivalent line lattices that we have obtained. The last column specifies whether there exist cases that admit non-invertible symmetries. Indeed, here we find that in some of the cases there exists a zero-form symmetry mapping some of the different line lattices, that are therefore equivalent. Furthermore in such cases we expect the existence of non-invertible symmetries obtained by combining the zero-form symmetry with a suitable gauging of the one-form symmetry.

A remarkable observation strengthening our results regards the fact that our analysis reproduces the limiting $G(k, k, 2)$ cases, where supersymmetry enhances to $\mathcal{N} = 4$ with $\mathfrak{su}(3)$, $\mathfrak{so}(5)$ and $\mathfrak{g}_2$ gauge groups for $k = 3$, 4 and 6 respectively. Another check of our result is that it matches with the cases $G(3, 1, 1)$ and $G(3, 3, 3)$, where an $\mathcal{N} = 1$ Lagrangian picture has been worked out in [42].

**Note added**: When concluding this paper, the reference [43] appeared on arXiv. There, they study the classification of zero, one and two-form symmetries in $\mathcal{N} = 3$ $S$-fold SCFTs. Their analysis is holographic, along the lines of the construction of [44] for $\mathcal{N} = 4$ SYM. We have checked that our results are in agreement with their predictions.

## 2 Generalities

### 2.1 Global structures from the IR

The strategy adopted here, as already discussed in the introduction, is inspired by the one of [9]. The main difference is that instead of using BPS quivers, not yet available for our $S$-folds, we take advantage of the type IIB geometric setups and probe the charge spectrum with $(p,q)$-strings – the bound state of $p$ fundamental strings F1 and $q$ Dirichlet strings D1.[1]

Despite this difference, the rest of the procedure is the one of [9] which we now summarize. Denote as

$$\gamma^i = \left( e_1^{(i)}, m_1^{(i)}; \ldots; e_r^{(i)}, m_r^{(i)} \right), \tag{1}$$

a basis vector of the electromagnetic lattice of dynamical state charges under the $U(1)_e^r \times U(1)_m^r$ gauge symmetry on the Coulomb branch. The spectrum of lines can be determined by considering a general line $\mathcal{L}$ with charge

$$\ell = \left( e_1^{(l)}, m_1^{(l)}; \ldots; e_r^{(l)}, m_r^{(l)} \right). \tag{2}$$

This is a genuine line operator if the Dirac pairings with all dynamical states $\Psi$ are integer:

$$\langle \Psi, \mathcal{L} \rangle \in \mathbb{Z}, \qquad \forall \Psi. \tag{3}$$

This can be rephrased as the condition

$$\sum_{j=1}^{r} e_j^{(i)} m_j^{(l)} - m_j^{(i)} e_j^{(l)} \in \mathbb{Z}, \qquad \forall i. \tag{4}$$

Furthermore, inserting a local operator with charge $\gamma_i$ on the worldline of a line with charge $\ell$ shifts its charge by $\gamma_i$. Therefore if a line with charge $\ell$ appears in the spectrum then a line with charges $\ell + \sum k_i \gamma_i$ with $k_i \in \mathbb{Z}$ must also appear. When classifying the spectrum of charges of the line operators of a QFT it is then useful to consider the charges $\ell$ modulo these insertions of local states. This gives rise to equivalence classes of charges with respect to the relation:

$$\ell \sim \ell + \gamma_i, \qquad \forall i. \tag{5}$$

Borrowing the nomenclature of [9], we will refer to such identification as screening and we will work with each equivalence class by picking one representative. The genuine lines after screening form a lattice. In general two such lines are not mutually local and a choice of global structure corresponds to a choice of a maximal sublattice of mutually local lines.

### 2.2 Charged states in $S_{k,l}$-folds

We aim to determine the electromagnetic charges of the local states generated by $(p,q)$-strings stretched between (images of) D3-branes in presence of an $S$-fold. The $S$-fold background of Type IIB string theory consist of a spacetime $\mathbb{R}^4 \times (\mathbb{R}^6/\mathbb{Z}_k)$ where the $\mathbb{Z}_k$ quotient involves an S-duality twist by an element $\rho_k \in SL(2,\mathbb{Z})$ of order $k$, where $k = 2,3,4,6$. For $k > 2$ the value of the axio-dilaton vev is fixed by the requirement that it must be invariant under the modular transformation associated to $\rho_k$. The matrices $\rho_k$ and the corresponding values[2] of $\tau$ are given in Table 2.

---

[1] In order to provide the IR spectrum of line operators of the SCFTs from this UV perspective, we assume the absence of wall-crossing. While such an assumption is *a priori* motivated by the high degree of supersymmetry, *a posteriori* it is justified by the consistency of our results with the literature.

[2] In our convention, an $SL(2,\mathbb{Z})$ transformation of the axio-dilaton $\tau \to (a\tau + b)/(c\tau + d)$ relates to a matrix $\rho_k = \begin{pmatrix} d & c \\ b & a \end{pmatrix}$. We also have $S = \begin{pmatrix} 0 & -1 \\ 1 & 0 \end{pmatrix}$ and $T = \begin{pmatrix} 1 & 0 \\ 1 & 1 \end{pmatrix}$.

Table 2: Elements $\rho_k$ of $SL(2,\mathbb{Z})$ of order $k$ used in $S$-fold projections, and the corresponding fixed coupling $\tau$.

| $SL(2,\mathbb{Z})$ | $S^2 = -\mathbb{I}_2$ | $(ST)^{-1}$ | $S$ | $(S^3T)^{-1}$ |
|---|---|---|---|---|
| $k$ | 2 | 3 | 4 | 6 |
| $\rho_k$ | $\begin{pmatrix} -1 & 0 \\ 0 & -1 \end{pmatrix}$ | $\begin{pmatrix} 0 & 1 \\ -1 & -1 \end{pmatrix}$ | $\begin{pmatrix} 0 & -1 \\ 1 & 0 \end{pmatrix}$ | $\begin{pmatrix} 0 & -1 \\ 1 & 1 \end{pmatrix}$ |
| $\rho_k^{-1}$ | $\begin{pmatrix} -1 & 0 \\ 0 & -1 \end{pmatrix}$ | $\begin{pmatrix} -1 & -1 \\ 1 & 0 \end{pmatrix}$ | $\begin{pmatrix} 0 & 1 \\ -1 & 0 \end{pmatrix}$ | $\begin{pmatrix} 1 & 1 \\ -1 & 0 \end{pmatrix}$ |
| $\tau$ | any $\tau$ | $e^{2i\pi/3}$ | $i$ | $e^{2i\pi/3}$ |

A stack of $N$ D3-branes probing the singular point of the $S$-fold background engineer an $\mathcal{N} = 3$ field theory on the worldvolume of the stack of D3-branes. It is useful to consider the $k$-fold cover of spacetime, and visualize the $N$ D3-branes together with their $(k-1)N$ images under the $S_k$-fold projection. We are going to label the $m$-th image of the $i$-th D3-brane with the index $i_m$, where $i = 1, \ldots, N$ and $m = 1, \ldots, k$.

Under the $S$-fold projection, the two-form gauge fields of the closed string sector $B_2$ and $C_2$ transform in the fundamental representation:

$$\begin{pmatrix} B_2 \\ C_2 \end{pmatrix} \to \rho_k \begin{pmatrix} B_2 \\ C_2 \end{pmatrix}. \tag{6}$$

Consistently, the $(p,q)$ strings charged under these potentials are mapped to $(p',q')$ where:

$$(p'\ q') = (p\ q) \cdot \rho_k^{-1}. \tag{7}$$

We denote a state associated to a $(p,q)$ connecting the $i_m$-th D3-brane and the $j_n$ D3-brane as:

$$|p,q\rangle_{i_m,j_n} = |-p,-q\rangle_{j_n,i_m}, \tag{8}$$

where we identity states with both opposite charges and orientation.

First, strings linking branes in the same copy of $\mathbb{R}^6/\mathbb{Z}_2$ transform as follows:

$$|p,q\rangle_{i_m,j_m} \to \zeta_k^{-1} |p',q'\rangle_{i_{m+1},j_{m+1}}, \tag{9}$$

where $(p',q')$ are related to $(p,q)$ by (7) and $\zeta_k$ is the primitive $k$-th root of unity. These states always collectively give rise to a single state in the quotient theory, with charges:

$$D3_i D3_j \ : \ (0,0; \ldots; \overbrace{p,q}^{i\text{-th}}; \ldots; \overbrace{-p,-q}^{j\text{-th}}; \ldots; 0,0). \tag{10}$$

An important ingredient we need to add to our picture is the discrete torsion for $B_2$ and $C_2$ [23, 45]. In presence of such a discrete torsion, a string going from the $i_m$-th brane to the $j_{m+1}$-th brane should pick up an extra phase which depends only on its $(p,q)$-charge and the couple $(\theta_{NS}, \theta_{RR})$. More precisely, one expects that the $S$-fold action can be written as follows [46]:[3]

$$|p,q\rangle_{i_m j_{m+1}} \to \zeta_k^{-1} e^{2\pi i(p\theta_{NS} + q\theta_{RR})} |p',q'\rangle_{i_{m+1} j_{m+2}}, \tag{11}$$

---

[3]We thank Shani Meynet for pointing out [46] to us.

Table 3: Different discrete torsions on O3-planes.

| O3-planes | $O3^-$ | $O3^+$ | $\widetilde{O3}^-$ | $\widetilde{O3}^+$ |
|---|---|---|---|---|
| $(\theta_{\mathrm{NS}}, \theta_{\mathrm{RR}})$ | $(0,0)$ | $(1/2,0)$ | $(0,1/2)$ | $(1/2,1/2)$ |

where again $(p', q')$ are related to $(p,q)$ by (7). For $i \neq j$, this always leads to the following state in the projected theory [47, 48]:[4]

$$D3_i D3_j^\rho : (0,0;\ldots;\overbrace{p,q}^{i\text{-th}};\ldots;\overbrace{-(p\,q)\cdot\rho_k}^{j\text{-th}};\ldots;0,0). \tag{12}$$

Note that this is the only case that might not lead to any state in the quotient theory when $i = j$, i.e. when a string links a brane and its image. When the quotient state exists, it has charges

$$D3_i D3_i^\rho : (0,0;\ldots;\overbrace{(p\,q)-(p\,q)\cdot\rho_k}^{i\text{-th}};\ldots;0,0). \tag{13}$$

Analogously, strings twisting around the $S$-fold locus $n$-times pick up $n$-times the phase in (11).

A last remark is that discrete torsion allows some strings to attach to the $S$-fold if the latter has the appropriate NS and/or RR charge. If this is the case, the state is mapped as in (9):

$$|p,q\rangle_{S_k i_m} \to |p',q'\rangle_{S_k i_{m+1}}, \tag{14}$$

and provides the following charge in the projected theory:

$$S_k D3_i : (0,0;\ldots;\overbrace{p,q}^{i\text{-th}};\ldots;0,0). \tag{15}$$

These rules are illustrated and details on discrete torsion are provided in the remaining of this section for orientifolds and $S$-folds separately.

**The case with $k = 2$: Orientifolds**

In this subsection we apply the formalism described above for orientifolds and reproduce the spectrum of strings known in the literature.

The matrix $\rho_2$ is diagonal, therefore the two $p$ and $q$ factors can be considered independently. In this case the field theory obtained after the projection is Lagrangian and can be studied in perturbative string theory with unoriented strings. Discrete torsion takes value in $(\theta_{\mathrm{NS}}, \theta_{\mathrm{RR}}) \in \mathbb{Z}_2 \oplus \mathbb{Z}_2$, giving four different choices of O3-planes related by $SL(2, \mathbb{Z})$ actions [45], see Table 3.

The orientifold action is then recovered from (9) and (11) with $\zeta_2 = -1$. First, we have

$$|p,q\rangle_{i_1 j_1} \to -|-p,-q\rangle_{i_2 j_2} = -|p,q\rangle_{j_2 i_2}. \tag{16}$$

For the strings that stretch from one fundamental domain of $\mathbb{R}^6/\mathbb{Z}_2$ to the next, there are four cases depending on the values of $\theta_{\mathrm{NS}}$ and $\theta_{\mathrm{RR}}$:

$$\begin{aligned}
O3^- &: & |p,q\rangle_{i_1 j_2} &\to -|p,q\rangle_{j_1 i_2}, \\
O3^+ &: & |p,q\rangle_{i_1 j_2} &\to -e^{p\pi i}|p,q\rangle_{j_1 i_2}, \\
\widetilde{O3}^- &: & |p,q\rangle_{i_1 j_2} &\to -e^{q\pi i}|p,q\rangle_{j_1 i_2}, \\
\widetilde{O3}^+ &: & |p,q\rangle_{i_1 j_2} &\to -e^{(p+q)\pi i}|p,q\rangle_{j_1 i_2}.
\end{aligned} \tag{17}$$

---

[4]The action on $(p,q)$ involves $\rho_k^{-1}$, see (7). In writing (12) however, we measure the charge with respect to the brane in the chosen fundamental domain, hence the appearance of $\rho_k$ instead of its inverse.

It is interesting to consider strings connecting one brane to its image, $i = j$. In the case of trivial discrete torsion, corresponding to the $O3^-$-plane, all such strings are projected out. On the contrary, in the $O3^+$ case, an F1-string linking mirror branes survives the projection, while a D1-string similarly positioned is projected out. We also find strings that can attach to the different orientifolds following [47]:

$$O3^- : \text{none}, \quad O3^+ : |0,1\rangle_{O3^+ i_m}, \quad \widetilde{O3}^- : |1,0\rangle_{\widetilde{O3}^- i_m}, \quad \widetilde{O3}^+ : |1,1\rangle_{\widetilde{O3}^+ i_m}, \tag{18}$$

as well as bound states of these.

**The cases with $k > 2$: $S$-folds**

The construction discussed above can be applied to $S_{k>2}$ in order to obtain the string states in the quotient theory. For $k > 2$, the discrete torsion groups have been computed in [23], the result being $\theta_{NS} = \theta_{RR} \in \mathbb{Z}_3$ for the $S_3$-case and $\theta_{NS} = \theta_{RR} \in \mathbb{Z}_2$ for the $S_4$-case. The $S_6$-fold does not admit non-trivial discrete torsion. It was also pointed out that, for the $S_3$-case, the choices $\theta_{NS} = \theta_{RR} = 1/3$ and $\theta_{NS} = \theta_{RR} = 2/3$ are related by charge conjugation; therefore everything boils down to whether the discrete torsion is trivial or not. Following the notation of [23], we denote as $S_{k,1}$ the $S$-folds with trivial discrete torsion and as $S_{k,k}$ the $S$-folds with non-trivial discrete torsion.

As before, the only states that might not lead to any state in the quotient theory are the strings linking different covers of $\mathbb{R}^6/\mathbb{Z}_k$. Equation (11) generalizes in the following way [46]: a state $|p,q\rangle_{i_m j_n}$ is mapped to $\zeta_k^{-1} e^{2\pi i(p\theta_{NS} + q\theta_{RR})} |p',q'\rangle_{i_{m+1} j_{n+1}}$ with $(p',q')$ obtained as in Equation (7). In more details:

$$\begin{aligned}
S_{3,1} &: \quad |p,q\rangle_{i_1 j_{m+1}} \rightarrow e^{-i2\pi/3} |q-p,-p\rangle_{i_2 j_{m+2}}, \\
S_{3,3} &: \quad |p,q\rangle_{i_1 j_{m+1}} \rightarrow e^{-i2\pi/3} e^{im(p+q)2\pi/3} |q-p,-p\rangle_{i_2 j_{m+2}}, \\
S_{4,1} &: \quad |p,q\rangle_{i_1 j_{m+1}} \rightarrow e^{-i\pi/2} |-q,p\rangle_{i_2 j_{m+2}}, \\
S_{4,4} &: \quad |p,q\rangle_{i_1 j_{m+1}} \rightarrow e^{-i\pi/2} e^{im(p+q)\pi} |-q,p\rangle_{i_2 j_{m+2}}, \\
S_{6,1} &: \quad |p,q\rangle_{i_1 j_{m+1}} \rightarrow e^{-i\pi/3} |p-q,p\rangle_{i_2 j_{m+2}}.
\end{aligned} \tag{19}$$

This shows that no state is projected out for $S_{3,1}$ and $S_{3,3}$. Analogously to the orientifold cases, we project out some strings linking mirror branes: $|p,q\rangle_{i_n i_{n+2}}$ in $S_{4,1}$ and $S_{4,4}$, and $|p,q\rangle_{i_n i_{n+3}}$ in $S_{6,1}$ respectively.

Finally, we get extra strings linking the $S$-fold to D-branes for the cases with discrete torsion. Following the discussion in [48], we know that these $S$-folds admit all kinds of $p$ and $q$ numbers:

$$S_{3,3} : |p,q\rangle_{S_{3,3} i_n}, \qquad S_{4,4} : |p,q\rangle_{S_{4,4} i_n}. \tag{20}$$

## 2.3 Dirac pairing from $(p,q)$-strings

Having determined the states associated to $(p,q)$-strings that survive the $S$-fold projection we now analyze the electromagnetic charges of these states. It is useful to consider the system of a stack of D3-branes and an $S_{k,\ell}$-fold on a generic point of the Coulomb branch. This corresponds to moving away the D3-branes from the S-plane. On a generic point of the Coulomb branch, the low energy theory on the D3-branes is a $U(1)_i^N$ gauge symmetry, where each $U(1)_i$ factor is associated to the $i$-th D3-brane. The theory includes massive charged states generated by the $(p,q)$-strings studied in the previous section. A $(p,q)$-string stretched between the $i$-th and $j$-th D3-brane has electric charge $p$ and magnetic charge $q$ under $U(1)_i$ as well as electric charge $-p$ and magnetic charge $-q$ under $U(1)_j$, and is neutral with respect to other branes. We organize the charges under the various $U(1)$s in a vector:

$$(e_1, m_1; e_2, m_2; \ldots; e_N, m_N), \tag{21}$$

where $e_i$ and $m_i$ are the electric and magnetic charge under $U(1)_i$, respectively. In this notation the charge of a string stretched between the $i$-th and $j$-th D3-brane in the same cover of $\mathbb{R}^6/\mathbb{Z}_2$ has charge:

$$D3_i D3_j : (0,0;\ldots;\overbrace{p,q}^{i-th};0,0;\ldots;\overbrace{-p,-q}^{j-th};\ldots), \tag{22}$$

where the dots stand for null entries. We will keep using this notation in the rest of the paper. A $(p,q)$-string stretched between the $i$-th D3-brane and the $l$-th image of the $j$-th D3-brane imparts electromagnetic charges $(p,q)$ under $U(1)_i$ and charges $-(p,q)\rho_k^l$ under $U(1)_j$. In formulas:

$$D3_i D3_j^{\rho^l} : (0,0;\ldots;\overbrace{p,q}^{i-th};0,0;\ldots;\overbrace{-(p\ q)\cdot\rho_k^l}^{j-th};\ldots). \tag{23}$$

The last ingredient for our analysis is given by the Dirac pairing between two states. Consider a state $\Psi$ with charges $e_i, m_i$ under $U(1)_i$ and a state $\Psi'$ with charges $e_i', m_i'$ under $U(1)_i$. The pairing between F1 and D1-strings in Type IIB dictates that the Dirac pairing between these states is given by:

$$\langle\Psi,\Psi'\rangle = \sum_{i=1}^{N}\left(e_i m_i' - m_i e_i'\right). \tag{24}$$

By using this construction we can reproduce the usual Dirac pairing of $\mathcal{N} = 4$ SYM with $ABCD$ gauge algebras. As an example we now reproduce the Dirac pairing of $D_N$, engineered as a stack of $N$ D3-branes probing an $O3^-$-plane. In this case the allowed $(p,q)$-strings have the following charges:

$$D3_i D3_j : (0,0;\ldots;\overbrace{p,q}^{i-th};0,0;\ldots;\overbrace{-p,-q}^{j-th};\ldots),$$
$$D3_i D3_j^{\rho} : (0,0;\ldots;\overbrace{p,q}^{i-th};0,0;\ldots;\overbrace{p,q}^{j-th};\ldots). \tag{25}$$

The states associated to $(1,0)$-strings correspond to the $\mathcal{W}$ bosons while the states associated to $(0,1)$-strings correspond to magnetic monopoles $\mathcal{M}$. For each root $\mathcal{W}_i$ of $D_N$ let $\mathcal{M}_i$ be the corresponding coroot. More precisely if $\mathcal{W}_i$ is associated to a $(1,0)$-string connecting two D3-branes, then the coroot $\mathcal{M}_i$ corresponds to the string $(0,1)$ stretched between the same pair of D3-branes. The only non-vanishing Dirac pairing is the one between a $\mathcal{W}_i$ boson and an $\mathcal{M}_j$ monopole. This pairing between the simple (co)roots $\mathcal{W}_i$ and $\mathcal{M}_j$ is given by the intersection between $\mathcal{W}_i$ and $\mathcal{W}_j$, explicitly:

$$\langle\mathcal{W}_i,\mathcal{M}_j\rangle = (A_{D_N})_{i,j}, \tag{26}$$

where $A_{D_N}$ is the Cartan matrix of the $D_N$ algebra, corresponding to an $\mathfrak{so}(2N)$ gauge theory. Indeed the intersection between F1 strings in the background of an $O3^-$ reproduces the intersection of the roots of $D_N$. The Dirac pairing (26) reproduces the Dirac pairing of $\mathfrak{so}(2N)$ $\mathcal{N} = 4$ SYM. Similar constructions for $O3^+$, $\widetilde{O3}^-$, and $\widetilde{O3}^+$ lead to the $B$ and $C$ cases (while branes in absence of orientifold would give $A$). The corresponding gauge algebras are summarized in Table 4.

## 2.4 Lines in O3-planes

Before moving to new results, we illustrate our method with well understood O3-planes. Specifically, we consider placing $N = 2$ D3-branes in the background of an $O3^+$-plane.

In this specific example, the F1-strings corresponding to elementary dynamical states in the quotient theory can be chosen to be $|1,0\rangle_{1_2 1_1}$ and $|1,0\rangle_{1_1 2_1}$. The first links the $i = 1$ brane to its mirror $(D3_1^{\rho} D3_1)$ and the second links the $i = 1$ to the $i = 2$ brane $(D3_1 D3_2)$. A pictorial

Table 4: F1-string, D1-string, and the F1-D1 bound state providing respectively the electric, magnetic, and dyonic charges of the projected $\mathcal{N} = 4$ gauge theory.

| O3-planes | F1-string | D1-string | F1-D1 bound state |
|---|---|---|---|
| $O3^-$ | $\mathfrak{so}(2N)$ | $\mathfrak{so}(2N)$ | $\mathfrak{so}(2N)$ |
| $O3^+$ | $\mathfrak{usp}(2N)$ | $\mathfrak{so}(2N+1)$ | $\mathfrak{usp}(2N)$ |
| $\widetilde{O3}^-$ | $\mathfrak{so}(2N+1)$ | $\mathfrak{usp}(2N)$ | $\mathfrak{usp}(2N)$ |
| $\widetilde{O3}^+$ | $\mathfrak{usp}(2N)$ | $\mathfrak{usp}(2N)$ | $\mathfrak{so}(2N+1)$ |

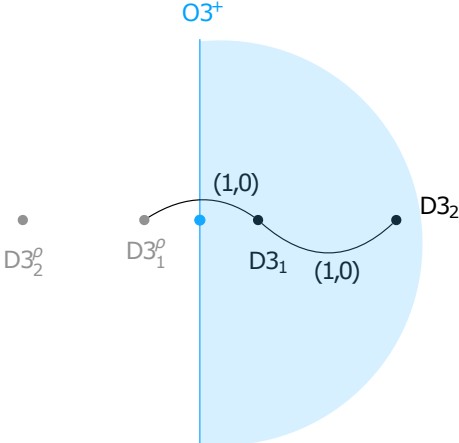

Figure 1: A pictorial representation of two D3-branes probing the $O3^+$ orientifold on a generic point of the Coulomb branch. The light blue shaded area is a possible choice of fundamental domain under the spacetime identification induced by the orientifold. Black (gray) dots represent (images of) D3-branes. Black lines correspond to $(p, q)$-strings stretched between D3-branes. In particular, we drew $(p, q)$-strings generating the $\mathcal{W}$-bosons corresponding to simple roots $\mathcal{N} = 4$ $\mathfrak{usp}(4)$ SYM.

representation of this setup is shown in Figure 1. In the notation of the previous section, they lead to $\mathcal{W}_i$-bosons in the gauge theory with the following charge basis:

$$D3_1^\rho D3_1 \,:\, w_1 = (2, 0; 0, 0), \quad D3_1 D3_2 \,:\, w_2 = (-1, 0; 1, 0). \tag{27}$$

These generate the algebra $\mathfrak{usp}(4)$ of electric charges. The elementary magnetic monopoles $\mathcal{M}_i$ come from the D1-strings $|0, 1\rangle_{O3^+1_1}$ and $|0, 1\rangle_{1_1 2_1}$, and provide the following charges:

$$O3^+ D3_1 \,:\, m_1 = (0, 1; 0, 0), \quad D3_1 D3_2 \,:\, m_2 = (0, -1; 0, 1). \tag{28}$$

This generates the algebra $\mathfrak{so}(5)$ of magnetic charges. Finally, the elementary $(1, 1)$-strings leading to states in the quotient theory can be chosen to be $|1, 1\rangle_{1_2 1_1}$ and $|1, 1\rangle_{1_1 2_1}$, i.e. $D3_1^\rho D3_1$ and $D3_1 D3_2$ respectively. They provide dyons $\mathcal{D}_i$:

$$D3_1^\rho D3_1 \,:\, d_1 = (2, 2; 0, 0), \quad D3_1 D3_2 \,:\, d_2 = (-1, -1; 1, 1), \tag{29}$$

which reproduces an $\mathfrak{usp}(4)$ algebra. We will limit ourselves to considering the $\mathcal{W}$-bosons and magnetic monopoles $\mathcal{M}$. Indeed, they generate the full lattice of electromagnetic charges admissible in the orientifold theory. See that

$$d_1 = w_1 + 2m_1, \qquad d_2 = w_2 + m_2. \tag{30}$$

Clearly, all other allowed $(p,q)$-charges can be reconstructed in this way. The Dirac pairing between these elementary electromagnetic charges reads

$$
\begin{aligned}
\langle \mathcal{W}_1, \mathcal{W}_2 \rangle = \langle \mathcal{M}_1, \mathcal{M}_2 \rangle &= 0\,, \\
\langle \mathcal{M}_1, \mathcal{W}_2 \rangle &= 1\,, \\
\langle \mathcal{W}_1, \mathcal{M}_1 \rangle = \langle \mathcal{M}_2, \mathcal{W}_1 \rangle = \langle \mathcal{W}_2, \mathcal{M}_2 \rangle &= 2\,.
\end{aligned}
\tag{31}
$$

Now, introduce a line operator $\mathcal{L}$ with charge vector $\ell$. It is convenient to express it in the basis of dynamical charges:

$$
\ell = \alpha_1 w_1 + \alpha_2 w_2 + \beta_1 m_1 + \beta_2 m_2\,,
\tag{32}
$$

where $\alpha_i$ and $\beta_i$ to be determined. Screening with respect to $\mathcal{W}_1$ and $\mathcal{W}_2$ imposes

$$
\alpha_1 \sim \alpha_1 + 1\,, \qquad \alpha_2 \sim \alpha_2 + 1\,,
\tag{33}
$$

respectively, while screening with respect to $\mathcal{M}_1$ and $\mathcal{M}_2$ imposes

$$
\beta_1 \sim \beta_1 + 1\,, \qquad \beta_2 \sim \beta_2 + 1\,.
\tag{34}
$$

Mutual locality with respect to the dynamical charges requires the quantities

$$
\begin{aligned}
\langle \mathcal{L}, \mathcal{W}_1 \rangle = -2\beta_1 + 2\beta_2\,, &\qquad \langle \mathcal{L}, \mathcal{W}_2 \rangle = \beta_1 - 2\beta_2\,, \\
\langle \mathcal{L}, \mathcal{M}_1 \rangle = 2\alpha_1 - \alpha_2\,, &\qquad \langle \mathcal{L}, \mathcal{M}_2 \rangle = -2\alpha_1 + 2\alpha_2\,,
\end{aligned}
\tag{35}
$$

to be integers. All these constraints set

$$
\alpha_1 = \frac{e}{2}\,, \quad \alpha_2 = 0\,, \quad \beta_1 = 0\,, \quad \beta_2 = \frac{m}{2} \qquad \mathrm{mod}\ 1\,,
\tag{36}
$$

with $e, m = 0, 1$. Linearity of the Dirac pairing then guarantees mutual locality with respect to the full dynamical spectrum. Thus, the charge of the most general line (modulo screening) must read:

$$
\ell_{e,m} = \frac{1}{2}(2e, -m; 0, m)\,.
\tag{37}
$$

A choice of global structure consists in finding a set of mutually local lines. The mutual locality condition between two lines $\mathcal{L}$ and $\mathcal{L}'$ with charges $\ell_{e,m}$ and $\ell_{e',m'}$ is given by:

$$
\langle \mathcal{L}, \mathcal{L}' \rangle = \frac{1}{2}(-em' + e'm) \in \mathbb{Z}\,.
\tag{38}
$$

Equivalently:

$$
em' - me' = 0 \quad \mathrm{mod}\ 2\,.
\tag{39}
$$

We find three such sets, each composed of a single line with non-trivial charge: $\ell_{1,0}$, $\ell_{0,1}$, or $\ell_{1,1}$. In agreement with [2], we find that the line with charge $\ell_{1,0}$ transforms as a vector of $\mathfrak{usp}(4)$ and the theory is $USp(4)$. The line with charge $\ell_{0,1}$ transforms as a spinor of $\mathfrak{so}(5)$ and corresponds to the global structure $(USp(4)/\mathbb{Z}_2)_0$. The line with charge $\ell_{1,1}$ transforms both as a vector and a spinor, and the gauge group is $(USp(4)/\mathbb{Z}_2)_1$. Motivated by the match between our results (obtained through the procedure described above) and the global structures of Lagrangian theories [2], in the next sections we use our method to analyze the line spectra of $S$-fold theories.

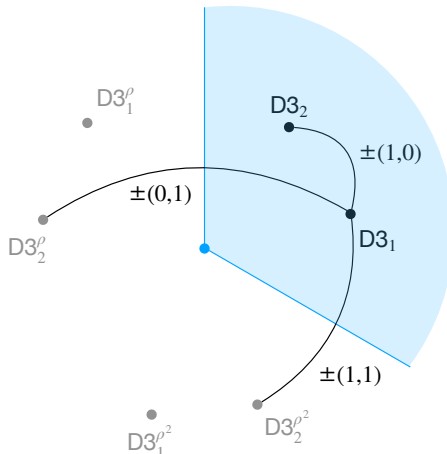

Figure 2: A pictorial representation of two D3-branes probing the $S_{3,1}$-fold. The transverse directions to the $S$-fold are shown. The light blue dot represents the position of the $S_{3,1}$-fold. The light blue shaded area is a possible choice of fundamental domain under the spacetime identification induced by the $S_{3,1}$-fold. Black (gray) dots represent (images of) D3-branes. Black lines correspond to $(p,q)$-strings stretched between D3-branes. In particular, we drew $(p,q)$-strings corresponding to $\mathcal{W}$-bosons of $\mathcal{N} = 4$ $\mathfrak{su}(3)$ SYM.

## 3 Lines in $S$-folds with $\mathcal{N} = 4$ enhancement

We now derive the spectrum of mutually local lines for the gauge theories obtained with $N = 2$ D3-branes in the background of an $S_{k,1}$ plane, in each case $k = 3$, 4 and 6. More precisely, exploiting the strategy spelled out in Section 2, we first compute the electromagnetic charge lattice of local states generated by $(p,q)$-strings. From this we extract the possible spectra of lines and compare them with the ones obtained in an $\mathcal{N} = 4$ Lagrangian formalism [2], since these theories have been claimed to enhance to $\mathcal{N} = 4$ SYM [49]. Matching the spectra provides an explicit dictionary between the various lattices and corroborates the validity of our procedure. In section 4 we will then generalize the analysis to the pure $\mathcal{N} = 3$ $S_{k,\ell}$ projections for any rank, thus providing the full classification for the one-form symmetries in all such cases.

### 3.1 Lines in $\mathfrak{su}(3)$ from $S_{3,1}$

**Dynamical states and their charges**

Two D3-branes probing the singular point of the $S_{3,1}$-fold are claimed to engineer $\mathfrak{su}(3)$ $\mathcal{N} = 4$ SYM. The charges of states generated by $(p,q)$-strings stretching between $D3_1$ and $D3_2$ or its first copy (see Figure 2) are

$$D3_1 D3_2 : (p,q;-p,-q), \quad D3_1 D3_2^\rho : (p,q;q,q-p), \quad D3_1 D3_2^{\rho^2} : (p,q;p-q,p). \quad (40)$$

One may also consider copies of the strings listed in Equation 40 such as:

$$D3_1^\rho D3_2^\rho : (-q,p-q;q,q-p), \quad (41)$$

as well as the strings going from one D3-brane to its own copies, for instance[5]

$$D3_1 D3_1^\rho : (2p-q,p+q;0,0). \quad (42)$$

---

[5]In the absence of discrete torsion, these states have not been considered previously in the literature [48, 50],

The charges of a generic string $D3_1D3_2^{\rho^2}$ in (40) can be expressed in terms of $D3_1D3_2$ and $D3_1D3_2^{\rho}$ charges:

$$D3_1D3_2^{\rho^2} : (p,q;p-q,p) = q(1,0;-1,0) + (q-p)(0,1;0,-1) \\ + (p-q)(1,0;0,-1) + p(0,1;1,1), \tag{43}$$

where the first two vectors on the RHS come from $D3_1D3_2$ with $p = 1$, $q = 0$ and $p = 0$, $q = 1$ respectively, and the last two come from $D3_1D3_2^{\rho}$ with $p = 1$, $q = 0$ and $p = 0$, $q = 1$ respectively. Acting with $\rho_3$, one can express all $D3_1^{\rho}D3_2^{\rho}$ and $D3_1^{\rho^2}D3_2^{\rho^2}$ charges in terms of $D3_1D3_2$ charges. The charges $D3_iD3_i^{\rho}$ can also be expressed as linear combinations of $D3_1D3_2^{\rho}$ and $D3_2^{\rho}D3_1^{\rho}$ charges. All in all, we find that the charges of the strings $D3_1D3_2$ and $D3_1D3_2^{\rho}$ form a basis of the lattice of dynamical charges.

The states corresponding to the $\mathcal{W}$-bosons generate the $\mathfrak{su}(3)$ algebra. One can take the strings $D3_1D3_2$ with $p = 1$ and $q = 0$ and $D3_1D3_2^{\rho}$ with $p = 0$ and $q = 1$ as representing a choice of positive simple roots. Their electromagnetic charge $w$ reads:

$$w_1 = (1,0;-1,0), \qquad w_2 = (0,1;1,1). \tag{44}$$

Furthermore, one can choose the strings $D3_1D3_2$ with $p = 0$ and $q = 1$ and $D3_1D3_2^{\rho}$ with $p = -1$ and $q = -1$ as generating the charge lattice of magnetic monopoles $\mathcal{M}$ of $\mathcal{N} = 4$ SYM with gauge algebra $\mathfrak{su}(3)$:

$$m_1 = (0,1;0,-1), \qquad m_2 = (-1,-1;-1,0). \tag{45}$$

The qualification of electric charges $\mathcal{W}$ and magnetic monopoles $\mathcal{M}$ of the $\mathcal{N} = 4$ theory makes sense since the Dirac pairing reads:

$$\langle \mathcal{W}_1, \mathcal{W}_2 \rangle = \langle \mathcal{M}_1, \mathcal{M}_2 \rangle = 0, \\ \langle \mathcal{W}_1, \mathcal{M}_1 \rangle = \langle \mathcal{W}_2, \mathcal{M}_2 \rangle = 2, \\ \langle \mathcal{W}_1, \mathcal{M}_2 \rangle = \langle \mathcal{W}_2, \mathcal{M}_1 \rangle = -1. \tag{46}$$

In [48, 50], it has been shown that these states correspond indeed to BPS states, and this is a strong check of the claim of the supersymmetry enhancement in this case.

**Line lattices**

Having identified the electromagnetic lattice of charges of $(p,q)$-strings we can now construct the spectrum of line operators and the corresponding one-form symmetries. It is useful to consider the charge $\ell = (e_1, m_1; e_2, m_2)$ of a general line $\mathcal{L}$ to be parameterized as follows:

$$\ell = \alpha_1 w_1 + \alpha_2 w_2 + \beta_1 m_1 + \beta_2 m_2 \\ = (\alpha_1 - \beta_2, \alpha_2 + \beta_1 - \beta_2; -\alpha_1 + \alpha_2 - \beta_2, \alpha_2 - \beta_1).$$

Screening with respect to $w_i$ and $m_i$ translates as the identifications:

$$\alpha_i \sim \alpha_i + 1, \qquad \beta_i \sim \beta_i + 1. \tag{47}$$

The Dirac pairing between the generic line $\mathcal{L}$ with charge $\ell$ given in (47) and the states $\mathcal{W}$ and $\mathcal{M}$ must be an integer, i.e.:

$$\langle \mathcal{L}, \mathcal{W}_1 \rangle = 2\beta_1 - \beta_2, \qquad \langle \mathcal{L}, \mathcal{W}_2 \rangle = -\beta_1 + 2\beta_2, \\ \langle \mathcal{L}, \mathcal{M}_1 \rangle = -2\alpha_1 + \alpha_2, \qquad \langle \mathcal{L}, \mathcal{M}_2 \rangle = \alpha_1 - 2\alpha_2 \qquad \in \mathbb{Z}. \tag{48}$$

---

and we do here for the sake of consistency with the analysis of section 2. Note however that since their charge (which is the only feature that matters in order to derive line spectra) can be expressed as linear combinations of the charges of more conventional states, our results are independent of whether we consider them or not.

Mutual locality with respect to the other states then follows by linearity as soon as (48) holds. Combining (47) and (48) we have

$$\alpha_1 = -\alpha_2 = \frac{e}{3}, \quad \text{and} \quad \beta_1 = -\beta_2 = \frac{m}{3},$$  (49)

for $e, m = 0, 1, 2$. Then, the charge of the most general line compatible with the spectrum of local operators modulo screening reads

$$\ell_{e,m} = \frac{1}{3}(2e - m, \ e + m; \ -e - m, \ e - 2m).$$  (50)

These charges form a finite $3 \times 3$ square lattice. The Dirac pairing between two lines $\mathcal{L}$ and $\mathcal{L}'$ with charges $\ell_{e,m}$ and $\ell_{e',m'}$ is

$$\langle \mathcal{L}, \mathcal{L}' \rangle = \frac{2}{3}(em' - e'm).$$  (51)

Two lines $\mathcal{L}$ and $\mathcal{L}'$ are mutually local if their Dirac pairing is properly quantized. In our conventions this corresponds to the requirement that $\langle \mathcal{L}, \mathcal{L}' \rangle$ is an integer:

$$e'm - em' = 0 \mod 3.$$  (52)

The lattice of lines together with the mutual locality condition obtained in (52) fully specifies the global structure of the $S_{3,1}$ SCFT of rank-2.

Our result is equivalent to the one obtained in [2] from the Lagrangian description of $\mathfrak{su}(3)$ $\mathcal{N} = 4$ SYM theory. Let us first write the charges in (50) as:

$$\ell_{e,m} = e \frac{w_1 - w_2}{3} + m \frac{m_1 - m_2}{3}.$$  (53)

Note that $(w_1 - w_2)/3$ (respectively, $(m_1 - m_2)/3$) is a weight of the electric (respectively, magnetic) algebra $\mathfrak{su}(3)$ with charge 1 under the center $\mathbb{Z}_3$ of the simply-connected group SU(3). Therefore, the line $\ell_{e,m}$ corresponds to a Wilson-'t Hooft line of charge $(e, m)$ under $\mathbb{Z}_3 \times \mathbb{Z}_3$.

As shown in [2], there are four possible lattices of mutually local Wilson-'t Hooft lines specified by two integers $i = 0, 1, 2$ and $p = 1, 3$. The corresponding gauge theories are denoted $(SU(3)/\mathbb{Z}_p)_i$ and relate to the line spectra we have obtained as follows:

$$
\begin{aligned}
SU(3) &\leftrightarrow \{\ell_{0,0}, \ell_{1,0}, \ell_{2,0}\}, \\
(SU(3)/\mathbb{Z}_3)_0 &\leftrightarrow \{\ell_{0,0}, \ell_{0,1}, \ell_{0,2}\}, \\
(SU(3)/\mathbb{Z}_3)_1 &\leftrightarrow \{\ell_{0,0}, \ell_{1,1}, \ell_{2,2}\}, \\
(SU(3)/\mathbb{Z}_3)_2 &\leftrightarrow \{\ell_{0,0}, \ell_{2,1}, \ell_{1,2}\}.
\end{aligned}
$$  (54)

It follows from linearity and screening that each lattice in the $S$-fold picture is determined by a single non-trivial representative, that can itself be identified by two integers $(e, m)$. For example, a possible choice is

$$(e, m) = (1, 0), \ (0, 1), \ (1, 1), \ (2, 1).$$  (55)

## 3.2  Lines in $\mathfrak{so}(5)$ from $S_{4,1}$

**Dynamical states and their charges**

Two D3-branes probing the singular point of the $S_{4,1}$-fold are claimed to engineer $\mathfrak{so}(5)$ $\mathcal{N} = 4$ SYM. Following a reasoning similar to one of the $S_{3,1}$-fold case, we can write all string charges as linear combinations of two kinds of strings, say

$$D3_1 D3_2 : (p, q; -p, -q), \quad D3_1 D3_2^\rho : (p, q; -q, p).$$  (56)

States corresponding to the $\mathcal{W}$-bosons of $\mathcal{N} = 4$ SYM are generated by $D3_1 D3_2$ with $p = 1$ and $q = 0$, and $D3_1 D3_2^\rho$ with $p = -1$ and $q = -1$. Their charges are

$$w_1 = (1, 0; -1, 0), \quad w_2 = (-1, -1; 1, -1). \tag{57}$$

These states generate the algebra $\mathfrak{so}(5)$ with short and long positive simple roots $w_1$ and $w_2$, respectively. A possible choice of states corresponding to elementary magnetic monopoles $\mathcal{M}$ is $D3_1 D3_2$ with $p = -1$ and $q = 1$, and $D3_1 D3_2^\rho$ with $p = 1$ and $q = 0$. The charges of these strings are:

$$m_1 = (-1, 1; 1, -1), \quad m_2 = (1, 0; 0, 1), \tag{58}$$

with $m_1$ the long and $m_2$ the short positive simple roots of the Langland dual algebra $\mathfrak{usp}(4)$. The Dirac pairings between $\mathcal{W}$ and $\mathcal{M}$ are as expected:

$$\begin{aligned}
\langle \mathcal{W}_1, \mathcal{W}_2 \rangle &= \langle \mathcal{M}_1, \mathcal{M}_2 \rangle = 0, \\
\langle \mathcal{W}_1, \mathcal{M}_1 \rangle &= \langle \mathcal{W}_2, \mathcal{M}_2 \rangle = \langle \mathcal{M}_1, \mathcal{W}_2 \rangle = 2, \\
\langle \mathcal{M}_2, \mathcal{W}_1 \rangle &= 1.
\end{aligned} \tag{59}$$

**Line lattices**

We begin by parametrizing the charge $\ell$ of a general line $\mathcal{L}$ as:

$$\begin{aligned}
\ell &= \alpha_1 w_1 + \alpha_2 w_2 + \beta_1 m_1 + \beta_2 m_2 \\
&= (\alpha_1 - \alpha_2 - \beta_1 + \beta_2, \beta_1 - \alpha_2; -\alpha_1 + \alpha_2 + \beta_1, -\alpha_2 - \beta_1 + \beta_2).
\end{aligned} \tag{60}$$

Screening with respect to the local states $\mathcal{W}$ and $\mathcal{M}$ translates as:

$$\alpha_i \sim \alpha_i + 1, \quad \beta_i \sim \beta_i + 1. \tag{61}$$

Mutual locality with respect to the dynamical states generated by $(p, q)$-strings reads:

$$\begin{aligned}
\langle \mathcal{L}, \mathcal{W}_1 \rangle &= 2\beta_1 - \beta_2, \\
\langle \mathcal{L}, \mathcal{W}_2 \rangle &= -2\beta_1 + 2\beta_2, \\
\langle \mathcal{L}, \mathcal{M}_1 \rangle &= -2\alpha_1 + 2\alpha_2, \\
\langle \mathcal{L}, \mathcal{M}_2 \rangle &= \alpha_1 - 2\alpha_2
\end{aligned} \quad \in \mathbb{Z}. \tag{62}$$

This imposes $\alpha_1 = \beta_2 = 0$ and $\alpha_2, \beta_1 \in \frac{1}{2}\mathbb{Z}$, and therefore the charge of the most general line compatible with the spectrum of local states can be written as:

$$\ell_{e,m} = \frac{e}{2} w_2 + \frac{m}{2} m_1 = \frac{1}{2}(-e - m, -e + m; e + m, -e - m). \tag{63}$$

The Dirac pairing between two lines $\mathcal{L}$ and $\mathcal{L}'$ with charges $\ell_{e,m}$ and $\ell_{e',m'}$ is:

$$\langle \mathcal{L}, \mathcal{L}' \rangle = \frac{1}{2}(e'm - em'). \tag{64}$$

Two such lines are mutually local if their Dirac pairing if $\langle \mathcal{L}, \mathcal{L}' \rangle$ is an integer, i.e.:

$$(e'm - em') = 0 \mod 2. \tag{65}$$

Therefore, the allowed lines form a finite $2 \times 2$ square lattice parametrized by $e, m = 0, 1$, where the mutual locality condition is given by (65). This reproduces the expected global structures of $\mathcal{N} = 4$ $\mathfrak{so}(5)$ SYM. There are three possible choices of maximal lattices of mutually local lines which correspond to the three possible global structures of $\mathfrak{so}(5)$. The explicit mapping

can be obtained by comparing the electromagnetic charges of the lines with the charges of the $\mathcal{W}$ bosons and monopoles $\mathcal{M}$, along the lines of the analysis of above in the $\mathfrak{su}(3)$ case. We obtain the following global structures:

$$
\begin{aligned}
Spin(5) &\leftrightarrow \{\ell_{0,0}, \ell_{1,0}\}, \\
SO(5)_0 &\leftrightarrow \{\ell_{0,0}, \ell_{0,1}\}, \\
SO(5)_1 &\leftrightarrow \{\ell_{0,0}, \ell_{1,1}\}.
\end{aligned}
\tag{66}
$$

### 3.3 Trivial line in $\mathfrak{g}_2$ from $S_{6,1}$

**Dynamical states and their charges**

Two $D3$-branes probing the singular point of the $S_{6,1}$-fold are claimed to engineer $\mathfrak{g}_2$ $\mathcal{N} = 4$ SYM. The charges of states generated by $(p, q)$-strings are:

$$
\begin{aligned}
D3_1 D3_2 &: (p,q;-p,-q), & D3_1 D3_2^{\rho} &: (p,q;-q,p-q), \\
D3_1 D3_2^{\rho^2} &: (p,q;p-q,p), & D3_1 D3_2^{\rho^3} &: (p,q;p,q), \\
D3_1 D3_2^{\rho^4} &: (p,q;q,-p+q), & D3_1 D3_2^{\rho^5} &: (p,q;-p+q,-p). \\
\text{etc.}
\end{aligned}
\tag{67}
$$

As shown in [23] and as before, one can choose a set of strings representing dynamical particles and generating the algebra $\mathfrak{g}_2$.

**Line lattice**

The analysis of the charge spectrum in the case of the $S_{6,1}$-fold can be carried out along the lines of the previous sections. One can show that the only line that is mutually local with respect to the local states generated by $(p, q)$-strings modulo screening is the trivial line with charges $\ell = (0, 0; 0, 0)$. This is consistent with the enhancement to $\mathcal{N} = 4$ with gauge algebra $\mathfrak{g}_2$ because the center of the simply-connected $G_2$ is trivial, which implies the absence of non-trivial lines [2]. There is only one possible global structure, and the one-form symmetry is trivial.

## 4 Lines in $\mathcal{N} = 3$ $S$-folds

In this section, we generalize the procedure spelled out in the previous sections to $S$-folds theories of arbitrary rank, and later to the cases with non-trivial discrete torsion for the $B_2$ and $C_2$ fields. This allows us to classify the line spectrum for every $\mathcal{N} = 3$ $S$-fold theory, and identify the one-form symmetry group as well as the allowed global structures for a given theory.

The basic ingredients needed in the analysis are the lattice of electromagnetic charges of local states and the Dirac pairing, both of which can be inferred from the Type IIB setup along the lines of the rank-2 cases studied in Section 3. As already emphasized, we work under the assumption that the states generated by $(p, q)$-string form a good set of representatives of the electromagnetic charge lattice of the full spectrum.

Note that it does not strictly make sense to talk about $(p, q)$-strings on the $\mathbb{R}^4 \times \mathbb{R}^6 / \mathbb{Z}_k$ $S$-fold background because the $S$-fold projection involves an $SL(2, \mathbb{Z})$ action which mixes F1 and D1 strings. This is analogous to the fact that in the orientifold cases it only makes sense to consider unoriented strings, since the orientifold action reverses the worldsheet parity (equivalently, it involves the element $-\mathbb{I}_2 \in SL(2, \mathbb{Z})$). Nevertheless it makes sense to consider oriented strings (together with their images) on the double cover of the spacetime; this allows the

computation of the electromagnetic charge lattice of local states and the Dirac pairing, as reviewed in Section 2. Similarly when dealing with $S_k$-folds we consider $(p,q)$-strings on the $k$-cover of the spacetime, and extract from this the charges of local states and the Dirac pairing. The spectrum of lines can then be obtained using the procedure of [9] reviewed in Section 2.

## 4.1 Lines in $S_{3,1}$-fold

Let us first determine the lattice of electromagnetic charges of dynamical states. The charges generated by $(p,q)$-strings on the background of an $S_{3,1}$ fold are given by

$$D3_i D3_j^{\rho^l} : (0,0;\ldots;\overbrace{p,q}^{i-th};\ldots;\overbrace{-(p\ q)\cdot \rho_3^l}^{j-th};\ldots;0,0).\tag{68}$$

This expression is obtained from a $(p,q)$-string stretched between the $i$-th D3-brane and the $l$-th image of the $j$-th D3-brane. Recall that $\rho_3$ generates a $\mathbb{Z}_3$ subgroup of $SL(2,\mathbb{Z})$. A possible basis for the lattice of charges generated by $(p,q)$-strings is given by:

$$\begin{aligned}
w_1 &= (1,0;-1,0;\ldots),\\
w_2 &= (0,1;1,1;\ldots),\\
m_1 &= (0,1;0,-1;\ldots),\\
m_2 &= (-1,-1;-1,0;\ldots),\\
P_i &= (1,0;0,0;\ldots;\overbrace{-1,0}^{i-th};0,0;\ldots),\\
Q_i &= (0,1;0,0;\ldots;\overbrace{0,-1}^{i-th};0,0;\ldots),
\end{aligned}\tag{69}$$

where $w_i$ and $m_i$ are the charges of the corresponding states in the rank-2 case, with all other entries set to 0. Let $\mathcal{P}_i$ and $\mathcal{Q}_i$ be the states with charges $P_i$ and $Q_i$ respectively, for $i = 3,\ldots,N$. Note that when the rank is $N > 2$, it does not make sense to talk about $\mathcal{W}$-bosons and magnetic monopoles $\mathcal{M}$ since the pure $\mathcal{N}=3$ theories are inherently strongly coupled and do not admit a Lagrangian description. Nevertheless, we will denote $\mathcal{W}_i$ and $\mathcal{M}_i$ the states with charges $w_i$ and $m_i$ respectively, by analogy with the above.

The charge $\ell$ of a general line $\mathcal{L}$ can be written as the linear combination:

$$\ell = \alpha_1 w_1 + \alpha_2 w_2 + \beta_1 m_1 + \beta_2 m_2 + \sum_{i=3}^N (\delta_i P_i + \gamma_i Q_i).\tag{70}$$

Besides, screening translates into the identifications:

$$\alpha_i \sim \alpha_i + 1, \quad \beta_i \sim \beta_i + 1, \quad \delta_i \sim \delta_i + 1, \quad \gamma_i \sim \gamma_i + 1.\tag{71}$$

Let us now analyze the constraints imposed on this line given by mutual locality with respect to the dynamical states generated by $(p,q)$-strings. Our results are summarized in Table 5.

Consider the mutual locality conditions:

$$\langle \mathcal{L}, \mathcal{P}_i - \mathcal{P}_j \rangle = \delta_i - \delta_j \in \mathbb{Z} \quad \Rightarrow \quad \delta_i = \delta_j = \delta, \qquad i,j = 3,\ldots,N,\tag{72}$$

and

$$\langle \mathcal{L}, \mathcal{Q}_i - \mathcal{Q}_j \rangle = \gamma_j - \gamma_i \in \mathbb{Z} \quad \Rightarrow \quad \gamma_j = \gamma_i = \gamma, \qquad i,j = 3,\ldots,N.\tag{73}$$

Table 5: The charges of allowed lines in the $S_{3,1}$-fold theories. The charges $w_i, m_i, P$ and $Q$ are given in (69), and $r, s = 0, 1, 2$. The mutual locality condition for two lines with charges $\ell_{r,s}$ and $\ell_{r',s'}$ is $rs' - sr' = 0 \mod 3$.

| Rank | Line charge |
|---|---|
| $3n$ | $\ell_{r,s} = \dfrac{r}{3}w_1 + \dfrac{s}{3}w_2 - \dfrac{r}{3}m_1 - \dfrac{s}{3}m_2 + \dfrac{r+s}{3}(P-Q)$ |
| $3n+1$ | $\ell_{r,s} = \dfrac{r}{3}w_1 + \dfrac{r-s}{3}w_2 + \dfrac{s}{3}m_1 + \dfrac{r}{3}m_2 + \dfrac{r+s}{3}(P-Q)$ |
| $3n+2$ | $\ell_{r,s} = \dfrac{r}{3}w_1 - \dfrac{r}{3}w_2 + \dfrac{s}{3}m_1 - \dfrac{s}{3}m_2 - \dfrac{r+s}{3}(P-Q)$ |

Furthermore, there are dynamical states with charges:

$$
(0,0;\dots;\overbrace{1,-1}^{i-th};\dots) = (p,q;\dots;\overbrace{-p,-q}^{i-th};\dots)\Big|_{\substack{p=0\\q=1}} + (p,q;\dots;\overbrace{p-q,p}^{i-th};\dots)\Big|_{\substack{p=0\\q=-1}},
$$

$$
(0,0;\dots;\overbrace{2,1}^{i-th};\dots) = (p,q;\dots;\overbrace{-p,-q}^{i-th};\dots)\Big|_{\substack{p=-1\\q=0}} + (p,q;\dots;\overbrace{p-q,p}^{i-th};\dots)\Big|_{\substack{p=1\\q=0}}.
$$

(74)

Mutual locality with respect to these implies:

$$
\gamma = -\delta, \qquad \delta \in \frac{1}{3}\mathbb{Z}. \tag{75}
$$

Therefore, the charge of a general line can be rewritten as:

$$
\ell = \alpha_1 w_1 + \alpha_2 w_2 + \beta_1 m_1 + \beta_2 m_2 + \delta(P-Q), \tag{76}
$$

where

$$
\begin{aligned}
P &= \sum_{i=3}^{N} p_i = (N-2,0;0,0;-1,0;-1,0;\dots;-1,0), \\
Q &= \sum_{i=3}^{N} q_i = (0,N-2;0,0;0,-1;0,-1;\dots;0,-1).
\end{aligned}
\tag{77}
$$

In (77), we have modified our notation slightly since the dots $\dots$ now represent a sequence of pairs $(-1,0)$ and $(0,-1)$ for $P$ and $Q$ respectively. Mutual locality between the line $\mathcal{L}$ and the generators of the charge lattice of dynamical states imposes the following constraints:

$$
\begin{aligned}
\langle \mathcal{L}, \mathcal{P}_i \rangle &= (N-1)\delta - \alpha_2 - \beta_1 + \beta_2, \\
\langle \mathcal{L}, \mathcal{Q}_i \rangle &= (N-1)\delta + \alpha_1 - \beta_2, \\
\langle \mathcal{L}, \mathcal{W}_1 \rangle &= (N-2)\delta - 2\beta_1 + \beta_2, \\
\langle \mathcal{L}, \mathcal{W}_2 \rangle &= (N-2)\delta - 2\beta_2 + \beta_1, \\
\langle \mathcal{L}, \mathcal{M}_1 \rangle &= (N-2)\delta + 2\alpha_1 - \alpha_2, \\
\langle \mathcal{L}, \mathcal{M}_2 \rangle &= -2(N-2)\delta - \alpha_1 + 2\alpha_2
\end{aligned}
\qquad \in \mathbb{Z}.
\tag{78}
$$

One can compute the following:

$$\langle \mathcal{L}, \mathcal{W}_1 + 2\mathcal{W}_2 \rangle = 3(N-2)\delta - 3\beta_2 \qquad \in \mathbb{Z} \Rightarrow \beta_2 \in \frac{1}{3}\mathbb{Z},$$

$$\langle \mathcal{L}, \mathcal{M}_1 + 2\mathcal{M}_2 \rangle = -3\alpha_1 \qquad \in \mathbb{Z} \Rightarrow \alpha_1 \in \frac{1}{3}\mathbb{Z},$$

$$\langle \mathcal{L}, \mathcal{W}_1 - \mathcal{W}_2 \rangle = 3(\beta_2 - \beta_1) \qquad \in \mathbb{Z} \Rightarrow \beta_1 \in \frac{1}{3}\mathbb{Z}, \tag{79}$$

$$\langle \mathcal{L}, \mathcal{M}_1 - \mathcal{M}_2 \rangle = 3(N-2)\delta + 3(\alpha_1 - \alpha_2) \quad \in \mathbb{Z} \Rightarrow \alpha_2 \in \frac{1}{3}\mathbb{Z}.$$

In brief, we have found that $\alpha_i, \beta_i, \delta \in \frac{1}{3}\mathbb{Z}$. It is now useful to treat separately three cases, depending on the value of $N \bmod 3$. In all these cases we find that the lines modulo screening can be arranged in a finite $3 \times 3$ lattice, the one-form symmetry group is $\mathbb{Z}_3$ and there are four choices of global structure.

**Case $N = 3n$**

The mutual locality conditions in (78) can be written as:

$$\begin{aligned}
\langle \mathcal{L}, \mathcal{P}_i \rangle &= -\delta - \alpha_2 - \beta_1 + \beta_2, \\
\langle \mathcal{L}, \mathcal{Q}_i \rangle &= -\delta + \alpha_1 - \beta_2, \\
\langle \mathcal{L}, \mathcal{W}_1 \rangle &= \delta - 2\beta_1 + \beta_2, \\
\langle \mathcal{L}, \mathcal{W}_2 \rangle &= \delta - 2\beta_2 + \beta_1, \\
\langle \mathcal{L}, \mathcal{M}_1 \rangle &= \delta + 2\alpha_1 - \alpha_2, \\
\langle \mathcal{L}, \mathcal{M}_2 \rangle &= \delta - \alpha_1 + 2\alpha_2
\end{aligned} \qquad \in \mathbb{Z}. \tag{80}$$

One computes that:

$$\begin{aligned}
\langle \mathcal{L}, \mathcal{Q}_i + \mathcal{W}_1 \rangle &= \alpha_1 + \beta_1 & \Rightarrow \beta_1 = -\alpha_1, \\
\langle \mathcal{L}, \mathcal{P}_i + \mathcal{W}_2 \rangle &= -\alpha_2 - \beta_2 & \Rightarrow \beta_2 = -\alpha_2, \\
\langle \mathcal{L}, \mathcal{Q}_i \rangle &= -\delta + \alpha_1 + \alpha_2 & \Rightarrow \delta = \alpha_1 + \alpha_2,
\end{aligned} \tag{81}$$

and this implies:

$$\alpha_1 = -\beta_1 = \frac{r}{3}, \qquad \alpha_2 = -\beta_2 = \frac{s}{3}, \qquad \delta = \frac{r+s}{3}, \qquad r,s = 0,1,2. \tag{82}$$

Therefore the lines form a finite $3 \times 3$ lattice parametrized by $r$ and $s$. Mutual locality between two general lines $\mathcal{L}$ and $\mathcal{L}'$ with charges $\ell_{r,s}$ and $\ell_{r',s'}$ reads:

$$\langle \mathcal{L}, \mathcal{L}' \rangle = \frac{2}{3}(sr' - rs') \in \mathbb{Z}, \tag{83}$$

or equivalently:

$$sr' - rs' = 0 \quad \bmod 3. \tag{84}$$

There are four possible choices of maximal lattices of mutually local lines. As in the rank-2 case discussed in section 3, each lattice is uniquely identified by one of its element, or equivalently by the pair $(r,s)$ of one of its non-trivial elements:

$$(r,s) = \begin{cases}
(1,0) \leftrightarrow \{\ell_{0,0}, \ell_{1,0}, \ell_{2,0}\}, \\
(0,1) \leftrightarrow \{\ell_{0,0}, \ell_{0,1}, \ell_{0,2}\}, \\
(1,1) \leftrightarrow \{\ell_{0,0}, \ell_{1,1}, \ell_{2,2}\}, \\
(1,2) \leftrightarrow \{\ell_{0,0}, \ell_{1,2}, \ell_{2,1}\}.
\end{cases} \tag{85}$$

**Case $N = 3n + 1$**

In this case the mutual locality constraints (78) are:

$$
\begin{aligned}
\langle \mathcal{L}, \mathcal{P}_i \rangle &= -\alpha_2 - \beta_1 + \beta_2 , \\
\langle \mathcal{L}, \mathcal{Q}_i \rangle &= \alpha_1 - \beta_2 , \\
\langle \mathcal{L}, \mathcal{W}_1 \rangle &= -\delta - 2\beta_1 + \beta_2 , \\
\langle \mathcal{L}, \mathcal{W}_2 \rangle &= -\delta - 2\beta_2 + \beta_1 , \\
\langle \mathcal{L}, \mathcal{M}_1 \rangle &= -\delta + 2\alpha_1 - \alpha_2 , \\
\langle \mathcal{L}, \mathcal{M}_2 \rangle &= 2\delta - \alpha_1 + 2\alpha_2
\end{aligned}
\qquad \in \mathbb{Z} .
\tag{86}
$$

One computes that:

$$
\begin{aligned}
\alpha_2 &= \alpha_1 - \beta_1 , \\
\delta &= \alpha_1 + \beta_1 , \\
\alpha_1 &= \beta_2 .
\end{aligned}
\tag{87}
$$

Therefore the most general $\alpha_i, \beta_i$ and $\delta$ satisfy:

$$
\alpha_1 = \beta_2 = \frac{r}{3} , \qquad \beta_1 = \frac{s}{3} , \qquad \alpha_2 = \frac{r-s}{3} , \qquad \delta = \frac{r+s}{3} , \qquad r, s = 0, 1, 2 .
\tag{88}
$$

The lines again form a finite $3 \times 3$ lattice parametrized by $r$ and $s$. Mutual locality between two general lines $\mathcal{L}$ and $\mathcal{L}'$ with charges $\ell_{r,s}$ and $\ell_{r',s'}$ reads:

$$
\langle \mathcal{L}, \mathcal{L}' \rangle = \frac{1}{3}(sr' - rs') \in \mathbb{Z} ,
\tag{89}
$$

or equivalently:

$$
sr' - rs' = 0 \mod 3 .
\tag{90}
$$

Similarly to the case $N = 3n$ there are four possible choices of maximal lattices of mutually local lines that can be indexed by one of their element, or equivalently by $(r, s) = (1, 0)$, $(0, 1)$, $(1, 1)$, $(1, 2)$.

**Case $N = 3n + 2$**

In this case, the mutual locality constraints (78) are

$$
\begin{aligned}
\langle \mathcal{L}, \mathcal{P}_i \rangle &= \delta - \alpha_2 - \beta_1 + \beta_2 , \\
\langle \mathcal{L}, \mathcal{Q}_i \rangle &= \delta + \alpha_1 - \beta_2 , \\
\langle \mathcal{L}, \mathcal{W}_1 \rangle &= -2\beta_1 + \beta_2 = \beta_1 + \beta_2 , \\
\langle \mathcal{L}, \mathcal{W}_2 \rangle &= -2\beta_2 + \beta_1 , \\
\langle \mathcal{L}, \mathcal{M}_1 \rangle &= 2\alpha_1 - \alpha_2 = -\alpha_1 - \alpha_2 , \\
\langle \mathcal{L}, \mathcal{M}_2 \rangle &= -\alpha_1 + 2\alpha_2
\end{aligned}
\qquad \in \mathbb{Z} .
\tag{91}
$$

One can compute that the solution is given by

$$
\begin{aligned}
\beta_2 &= -\beta_1 , \\
\alpha_2 &= -\alpha_1 , \\
\delta &= -\alpha_1 - \beta_1 .
\end{aligned}
\tag{92}
$$

Therefore the most general $\alpha_i, \beta_i$ and $\delta$ satisfy:

$$
\alpha_1 = -\alpha_2 = \frac{r}{3} , \qquad \beta_1 = -\beta_2 = \frac{s}{3} , \qquad \delta = -\frac{r+s}{3} , \qquad r, s = 0, 1, 2 .
\tag{93}
$$

Dirac pairing between two general lines $\mathcal{L}$ and $\mathcal{L}'$ with charges $\ell_{r,s}$ and $\ell_{r',s'}$ reads:

$$\langle \mathcal{L}, \mathcal{L}' \rangle = \frac{2}{3} \left( sr' - rs' \right) \in \mathbb{Z} \,. \tag{94}$$

Two such lines are mutually local if they satisfy the constraint:

$$sr' - rs' = 0 \mod 3 \,. \tag{95}$$

As before, there are four possible choices of maximal lattices of mutually local lines that can be indexed by one of their element, or equivalently by

$$(r,s) = (1,0),\ (0,1),\ (1,1),\ (1,2) \,. \tag{96}$$

## 4.2 Lines in $S_{4,1}$-fold

We now study the spectrum of lines in theories engineered by a stack of D3-branes probing the $S_{4,1}$-fold. The charges of states generated by a $(p,q)$-string on the background of an $S_{4,1}$-fold read

$$D3_i D3_j^{\rho^l} : (0,0;\dots;\overbrace{p,q}^{i-th};\dots;\overbrace{-(p\ q)\cdot\rho_4^l}^{j-th};\dots;0,0) \,, \tag{97}$$

for a $(p,q)$-strings stretched between the $i$-th D3-brane and the $l$-th image of the $j$-th D3-brane. One possible basis for the lattice of charges generated by $(p,q)$-strings is:

$$\begin{aligned}
w_1 &= (1,0;-1,0;0,0;\dots) \,, \\
w_2 &= (-1,-1;1,-1;0,0;\dots) \,, \\
m_1 &= (-1,1;1,-1;0,0;\dots) \,, \\
m_2 &= (1,0;0,1;0,0;\dots) \,, \\
P_i &= (1,0;0,0;\dots;\overbrace{-1,0}^{i-th};0,0;\dots) \,, \\
Q_i &= (0,1;0,0;\dots;\overbrace{0,-1}^{i-th};0,0;\dots) \,,
\end{aligned} \tag{98}$$

where $w_i$ and $m_i$ are the charges of the corresponding states in the rank-2 case, with all other entries set to 0. We denote $\mathcal{W}_i, \mathcal{M}_i, \mathcal{P}_i$ and $\mathcal{Q}_i$ the states with charges $w_i$, $m_i$, $P_i$ and $Q_i$, respectively.

The charge $\ell$ of a general line $\mathcal{L}$ can be written as the linear combination:

$$\ell = \alpha_1 w_1 + \alpha_2 w_2 + \beta_1 m_1 + \beta_2 m_2 + \sum_{i=3}^{N} (\delta_i P_i + \gamma_i Q_i) \,. \tag{99}$$

Screening translates into the identifications:

$$\alpha_i \sim \alpha_i + 1 \,, \quad \beta_i \sim \beta_i + 1 \,, \quad \delta_i \sim \delta_i + 1 \,, \quad \gamma_i \sim \gamma_i + 1 \,. \tag{100}$$

In the remainder of this section we compute the constraints imposed by mutual locality between the general line $\mathcal{L}$ and dynamical states. Our results are summarized in Table 6.

Consider first the mutual locality conditions:

$$\langle \mathcal{L}, \mathcal{P}_i - \mathcal{P}_j \rangle = \delta_i - \delta_j \in \mathbb{Z} \quad \Rightarrow \quad \delta_i = \delta_j = \delta \,, \tag{101}$$

$$\langle \mathcal{L}, \mathcal{Q}_i - \mathcal{Q}_j \rangle = \gamma_j - \gamma_i \in \mathbb{Z} \quad \Rightarrow \quad \gamma_j = \gamma_i = \gamma \,. \tag{102}$$

Table 6: The charges of allowed lines in the $S_{4,1}$-fold theories. The charges $w_i, m_i, P$ and $Q$ are given in (98), (77), and $r, s = 0, 1$. The mutual locality condition for two lines with charges $\ell_{r,s}$ and $\ell_{r',s'}$ is $rs' - sr' = 0 \mod 2$.

| Rank | Line charge |
|------|-------------|
| $2n$ | $\ell_{r,s} = \dfrac{r}{2} w_2 + \dfrac{s}{2} m_1 + \dfrac{r+s}{2}(P - Q)$ |
| $2n+1$ | $\ell_{r,s} = \dfrac{r}{2} w_1 + \dfrac{s}{2} w_2 + \dfrac{s}{2} m_1 + \dfrac{r}{2} m_2 + \dfrac{r}{2}(P - Q)$ |

Furthermore, there are dynamical states with charges:

$$
\begin{aligned}
(0,0;\ldots;\overset{i-th}{\overbrace{1,-1}};\ldots) &= (p,q;\ldots;\overset{i-th}{\overbrace{-p,-q}};\ldots)\Big|_{\substack{p=0\\q=1}} + (p,q;\ldots;\overset{i-th}{\overbrace{-q,p}};\ldots)\Big|_{\substack{p=0\\q=-1}}, \\
(0,0;\ldots;\overset{i-th}{\overbrace{1,1}};\ldots) &= (p,q;\ldots;\overset{i-th}{\overbrace{-p,-q}};\ldots)\Big|_{\substack{p=-1\\q=0}} + (p,q;\ldots;\overset{i-th}{\overbrace{-q,p}};\ldots)\Big|_{\substack{p=1\\q=0}}.
\end{aligned}
\tag{103}
$$

and mutual locality with respect to these states implies:

$$
\gamma = -\delta, \qquad \delta \in \frac{1}{2}\mathbb{Z}.
\tag{104}
$$

Therefore, the charge of a general line can be rewritten as:

$$
\ell = \alpha_1 w_1 + \alpha_2 w_2 + \beta_1 m_1 + \beta_2 m_2 + \delta(P - Q),
\tag{105}
$$

where $P$ and $Q$ are defined in (77). Mutual locality between the line $\mathcal{L}$ and the generators of the charge lattice of dynamical states implies:

$$
\begin{aligned}
\langle \mathcal{L}, \mathcal{P}_i \rangle &= (N-1)\delta + \alpha_2 - \beta_1, \\
\langle \mathcal{L}, \mathcal{Q}_i \rangle &= (N-1)\delta + \alpha_1 - \alpha_2 - \beta_1 + \beta_2, \\
\langle \mathcal{L}, \mathcal{W}_1 \rangle &= (N-2)\delta - 2\beta_1 + \beta_2, \\
\langle \mathcal{L}, \mathcal{W}_2 \rangle &= 2(N-2)\delta - 2\beta_2 + 2\beta_1, \\
\langle \mathcal{L}, \mathcal{M}_1 \rangle &= 2\alpha_1 - 2\alpha_2, \\
\langle \mathcal{L}, \mathcal{M}_2 \rangle &= (N-2)\delta - \alpha_1 + 2\alpha_2
\end{aligned}
\qquad \in \mathbb{Z}.
\tag{106}
$$

One computes the following:

$$
\begin{aligned}
\langle \mathcal{L}, \mathcal{W}_1 + \mathcal{W}_2 - \mathcal{M}_1 - \mathcal{M}_2 \rangle &= -\beta_2 - \alpha_1 \in \mathbb{Z} &\Rightarrow \beta_2 = -\alpha_1, \\
\langle \mathcal{L}, \mathcal{Q}_i + \mathcal{P}_i \rangle &= -2\beta_1 \in \mathbb{Z} &\Rightarrow \beta_1 \in \frac{1}{2}\mathbb{Z}, \\
\langle \mathcal{L}, \mathcal{Q}_i - \mathcal{P}_i \rangle &= -2\alpha_2 \in \mathbb{Z} &\Rightarrow \alpha_2 \in \frac{1}{2}\mathbb{Z}, \\
\langle \mathcal{L}, \mathcal{M}_1 \rangle &= 2\alpha_1 \in \mathbb{Z} &\Rightarrow \alpha_1, \beta_2 \in \frac{1}{2}\mathbb{Z}.
\end{aligned}
\tag{107}
$$

We have thus shown that $\alpha_i, \beta_i, \delta \in \frac{1}{2}\mathbb{Z}$ and $\alpha_1 = -\beta_2$. It is now useful to treat separately the cases of odd and even $N$. In both cases we find that the lines form a $2 \times 2$ lattice, the one-form symmetry is $\mathbb{Z}_2$ and there are three choices of global structure.

**Case $N = 2n$**

Mutual locality conditions (106) read:

$$
\begin{aligned}
\langle \mathcal{L}, \mathcal{P}_i \rangle &= -\delta - \beta_1 + \alpha_2 \,, \\
\langle \mathcal{L}, \mathcal{Q}_i \rangle &= -\delta - \alpha_2 - \beta_1 \,, \\
\langle \mathcal{L}, \mathcal{W}_1 \rangle &= \beta_2 \,, \\
\langle \mathcal{L}, \mathcal{W}_2 \rangle &= 0 \,, \\
\langle \mathcal{L}, \mathcal{M}_1 \rangle &= 0 \,, \\
\langle \mathcal{L}, \mathcal{M}_2 \rangle &= -\alpha_1
\end{aligned}
\qquad \in \mathbb{Z} \,,
\tag{108}
$$

and each solution can be written as:

$$
\alpha_2 = \frac{r}{2} \,, \qquad \beta_1 = \frac{s}{2} \,, \qquad \alpha_1 = \beta_2 = 0 \,, \qquad \delta = \frac{r+s}{2} \,, \qquad r, s = 0, 1 \,.
\tag{109}
$$

Therefore the lines form a $2 \times 2$ lattice parametrized by $r, s$. Mutual locality between two lines $\mathcal{L}$ and $\mathcal{L}'$ with charges $\ell_{r,s}$ and $\ell_{r',s'}$ respectively translates into:

$$
\langle \mathcal{L}, \mathcal{L}' \rangle = \frac{1}{2}(r's - rs') \in \mathbb{Z} \,,
\tag{110}
$$

or equivalently:

$$
r's - rs' = 0 \mod 2 \,.
\tag{111}
$$

The one-form symmetry group is thus $\mathbb{Z}_2$ and there are three different choices of maximal lattices of mutually local lines parametrized by $(r,s) = (1,0), \ (0,1), \ (1,1)$.

**Case $N = 2n + 1$**

The Dirac pairings (106) read:

$$
\begin{aligned}
\langle \mathcal{L}, \mathcal{P}_i \rangle &= \alpha_2 - \beta_1 \,, \\
\langle \mathcal{L}, \mathcal{Q}_i \rangle &= -\alpha_2 - \beta_1 \,, \\
\langle \mathcal{L}, \mathcal{W}_1 \rangle &= \delta + \beta_2 \,, \\
\langle \mathcal{L}, \mathcal{W}_2 \rangle &= 0 \,, \\
\langle \mathcal{L}, \mathcal{M}_1 \rangle &= 0 \,, \\
\langle \mathcal{L}, \mathcal{M}_2 \rangle &= \delta - \alpha_1
\end{aligned}
\qquad \in \mathbb{Z} \,,
\tag{112}
$$

and the general solution can be written as:

$$
\alpha_1 = \beta_2 = \delta = \frac{r}{2} \,, \qquad \alpha_2 = \beta_1 = \frac{s}{2} \,, \qquad r, s = 0, 1 \,.
\tag{113}
$$

Mutual locality between two lines $\mathcal{L}$ and $\mathcal{L}'$ with charges $\ell_{r,s}$ and $\ell_{r',s'}$ respectively translates into:

$$
\langle \mathcal{L}, \mathcal{L}' \rangle = \frac{1}{2}(r's - rs') \in \mathbb{Z} \,,
\tag{114}
$$

or equivalently:

$$
r's - rs' = 0 \mod 2 \,.
\tag{115}
$$

As in the previous case, the one-form symmetry group is therefore $\mathbb{Z}_2$ and there are three different choices of maximal lattices of mutually local lines that can be parametrized by:

$$
(r,s) = (1,0), \ (0,1), \ (1,1) \,.
\tag{116}
$$

## 4.3 Trivial line in $S_{6,1}$-fold

The analysis of the spectrum of lines in the case of the $S_{6,1}$-fold can be carried out along the lines of the previous subsections. One finds that the integer lattice of charges associated to $(p,q)$-strings is fully occupied. To see this notice that there are two states with the following charges:

$$
\begin{aligned}
(1,0;0,0;0,0;\dots) &= (p,q;p-q,p;0,0;\dots)\Big|_{\substack{p=0\\q=-1}} - (p,q;-q,p,0,0;\dots)\Big|_{\substack{p=1\\q=0}}, \\
(0,1;0,0;0,0;\dots) &= (1,0;0,0;0,0;\dots) - (p,q;-p-q;0,0;\dots)\Big|_{\substack{p=0\\q=1}} \\
&\quad - (p,q;-q,p;0,0;\dots)\Big|_{\substack{p=0\\q=-1}}.
\end{aligned}
\tag{117}
$$

By combining these states with $\mathcal{P}_i$ and $\mathcal{Q}_i$ we can obtain states with electric or magnetic charge 1 with respect to the $i$-th brane, and all other charges set to zero. Let us now consider a general line $\mathcal{L}$ with charge $\ell = (e_1, m_1; e_2, m_2; \dots)$. Mutual locality with respect to the local states we have just discussed implies:

$$
e_i, m_i \in \mathbb{Z} \quad \forall\, i,
\tag{118}
$$

and the insertion of the same local states along the lines translates to the identification:

$$
e_i \sim e_i + 1, \qquad m_i \sim m_i + 1.
\tag{119}
$$

Therefore, the only allowed line modulo screening is the trivial line, with charge $\ell = (0,0;0,0;\dots)$. This implies that the one form symmetry group is trivial, and accordingly there is only one possible choice of global form.

## 4.4 Trivial line in the discrete torsion cases

We generalize the analysis discussed in the previous sections to the cases with non-trivial discrete torsion in the $S_{3,3}$-fold and $S_{4,4}$-fold.

As we argued in Section 2 all the strings states that are present when the discrete torsion is trivial are also allowed when the discrete torsion is non-zero. Furthermore, there are strings ending on the $S$-fold itself, as discussed in Section 2. Thus, the lattice of charges of local states in the case of the $S_{3,3}$-fold and $S_{4,4}$-fold are generated by strings stretched between (images of) D3-branes – as in the cases with trivial discrete torsion – together with those additional strings. One can show that the integer lattice of electromagnetic charges of dynamical states is then fully occupied. Therefore, by a similar argument to the one used in the case of the $S_{6,1}$-fold in Section 4.3, the only line that is allowed is the trivial one, and the one-form symmetry group is $\mathbb{1}$ for the $S_{3,3}$-fold and $S_{4,4}$-fold with non-zero discrete torsion.

# 5 Non-invertible symmetries

We now discuss the possible presence of non-invertible symmetries in $S$-fold theories. In the case of $\mathcal{N} = 4$ theories, the presence of S-duality orbits can imply the existence of non-invertible duality defects which are built by combining the action of some element of $SL(2,\mathbb{Z})$ and the gauging of a discrete one-form symmetry [51–62].

Similar structures can be inferred from the $S$-fold construction. Consider moving one of the D3-brane along the non-contractible one-cycle of $S^5/\mathbb{Z}_k$ until it reaches its original position. The brane configurations before and after this are identical, and therefore the $S$-fold theories are invariant under this action. Going around the non-contractible one-cycle of $S^5/\mathbb{Z}_k$ in the case an $S_{k,l}$-fold involves an $SL(2,\mathbb{Z})$-transformation on the electric and magnetic charges $e_i$,

$m_i$ associated to the D3-brane that has been moved. Let $\Sigma_k^i$ denote the process of moving the $i$-th D3-brane along the non-contractible cycle of an $S_{k,l}$-fold. The action of $\Sigma_k^i$ on the charges is:

$$\Sigma_k^i : \begin{pmatrix} e_j \\ m_j \end{pmatrix} \rightarrow \begin{cases} \rho_k \cdot \begin{pmatrix} e_j \\ m_j \end{pmatrix}, & j = i, \\[2mm] \begin{pmatrix} e_j \\ m_j \end{pmatrix}, & j \neq i. \end{cases} \tag{120}$$

The charge lattice of dynamical states is invariant under $\Sigma_k^i$, while the set of line lattices can be shuffled. Consider for example the $S_{3,1}$-case with rank $N = 2$. One can compute explicitly the following orbits:

$$(1,0) \longleftrightarrow (0,1) \longleftrightarrow (1,1) \qquad (1,2) \, \text{↻} \tag{121}$$

where the pairs $(e,m)$ parametrize the maximal sub-lattice of mutually local lines as discussed in section (3.1). Two line lattices connected by an arrow in (121) are mapped to each other under proper combinations of $\Sigma_3^i$.

This theory enhances to $\mathfrak{su}(3)$ $\mathcal{N} = 4$ SYM. Using the mapping (54) between the line lattices parametrized by $(e,m)$ and the global structures of $\mathfrak{su}(3)$, the formula (121) reproduces the $\mathcal{N} = 4$ orbits under the element $ST \in SL(2,\mathbb{Z})$. As shown in the literature [51, 52, 54, 55], this transformation can be combined with a proper gauging of the one-form symmetry to construct the non-invertible self-duality defects of $\mathfrak{su}(3)$ at $\tau = e^{2\pi i/3}$. Therefore in our notation we expect the existence of non-invertible symmetries involving $\Sigma_k^i$ for the lattices labeled by $(e,m) = (1,0),(0,1),(1,1)$, and none in the $(e,m) = (1,2)$ case.

Similarly, one can consider the orbits in the case of $S_{4,1}$ with $N = 2$, where the SCFT enhances to $\mathfrak{so}(5)$ $\mathcal{N} = 4$ SYM. By using the transformations $\Sigma_4^i$ as above we find the following orbits

$$(0,1) \longleftrightarrow (1,0) \qquad (1,1) \, \text{↻} \tag{122}$$

where the pairs $(e,m)$ parametrize the maximal sub-lattices of mutually local lines as discussed in section (3.2).

These reproduce the $\mathcal{N} = 4$ orbits under the element $S \in SL(2,\mathbb{Z})$. Again this transformation can be combined with a proper gauging of the one-form symmetry to construct the non-invertible self-duality defects of $\mathfrak{so}(5)$ at $\tau = i$.

Motivated by this match, one can expect that in the case of general rank, non-invertible symmetries will be present when multiple choices of maximal sub-lattices of mutually local lines are related by the transformations $\Sigma_k^i$, as above. The orbits are:

$$S_{3,1}: \quad (1,0) \longleftrightarrow (0,1) \longleftrightarrow (1,1) \quad (1,2) \, \text{⟲} \tag{123}$$

$$S_{4,1}: \begin{cases} (0,1) \longleftrightarrow (1,0) \quad (1,1) \, \text{⟲} & N = 0 \mod 2, \\[2mm] (1,0) \longleftrightarrow (1,1) \quad (0,1) \, \text{⟲} & N = 1 \mod 2, \end{cases} \tag{124}$$

where the pairs $(r,s)$ parametrize the maximal sub-lattices of mutually local lines as in section 4.

In the $S_{6,1}$, $S_{3,3}$ and $S_{4,4}$-cases, there is only one possible global structure that is mapped to itself by the $\Sigma_k^i$ transformations.

By analogy with the cases where there is $\mathcal{N} = 4$ enhancement, we expect the existence of non-invertible symmetries when the transformations $\Sigma_k^i$ map different line lattices, built by combining this $\Sigma_k^i$-action with a suitable gauging of the one-form symmetry.

## 6 Conclusions

In this paper, we have exploited the recipe of [9] for arranging the charge lattice of genuine lines modulo screening by dynamical particles. We have adapted such strategy, originally designed for BPS quivers, to the case of $(p, q)$-strings, in order to access to the electromagnetic charges of non-Lagrangian $\mathcal{N} = 3$ $S$-fold SCFTs. This procedure has allowed us to provide a full classification of the one-form symmetries of every $S$-fold SCFT. We singled out two cases with a non-trivial one-form symmetry, corresponding to the $\mathbb{Z}_3$ and the $\mathbb{Z}_4$ $S$-folds in absence of discrete torsion, denoted here as $S_{3,1}$ and $S_{4,1}$ respectively. Our results are consistent with the supersymmetry enhancement that takes place when two D3-branes are considered. Lastly, we discuss the possibility of non-invertible duality defects, by recovering the expected results for the cases with supersymmetry enhancement and proposing a generalization at any rank.

We left many open questions that deserve further investigations. It would for example be interesting to study in more details the projection of the states generated by the $(p, q)$-configurations in an $S$-fold background. In the present article, the only relevant information was the electromagnetic charges carried by these states, but a deeper analysis of the dynamics of these $S$-fold theories requires more work. This would in turn improve our understanding of their mass spectrum. For instance, a comparison of the BPS spectrum could be made exploiting the Lagrangian descriptions of [42]. This could also help finding the origin of the mapping between the multiple lattices found in the $S_{3,1}$ and $S_{4,1}$-cases. Further investigations in this direction would deepen our geometric understanding of the non-invertible symmetries expected in this class of theories, along the lines of the brane analysis of [63–65].

It would also be of interest to generalize the analysis to other $\mathcal{N} = 3$ SCFTs that are not constructed from $S$-fold projections, such as the exceptional $\mathcal{N} = 3$ theories [24, 30]. These theories can be obtained from M-theory backgrounds and one may study the charge lattice with probe M2-branes. One could therefore apply an analysis similar to the one spelled in [66–70]. Regarding the $S$-fold constructions, the cases of $S$-folds with $\mathcal{N} = 2$ supersymmetry [71, 72] also deserve further investigations (see [73, 74] for similar analysis in class $S$ theories). In the absence of BPS quivers, one needs to adapt the UV analysis of [9]. In general, one would like to find a stringy description that avoids wall crossing and allows reading the charge lattices and the one-form symmetries for such theories.

## Acknowledgements

We are grateful to Iñaki García Etxebarria for valuable insights on the manuscript, and to Shani Meynet and Robert Moscrop for useful discussions. A.P. thanks the Theoretical and Mathematical Physics group from Université Libre de Bruxelles for their hospitality during the completion of this work.

**Funding information** The work of A.A., D.M., A.P. and S.R. has been supported in part by the Italian Ministero dell'Istruzione, Università e Ricerca (MIUR), in part by Istituto Nazionale di Fisica Nucleare (INFN) through the "Gauge Theories, Strings, Supergravity" (GSS) research project and in part by MIUR-PRIN contract 2017CC72MK-003. V.T. acknowledges funding by the Deutsche Forschungsgemeinschaft (DFG, German Research Foundation) under Germany's Excellence Strategy EXC 2181/1 - 390900948 (the Heidelberg STRUCTURES Excellence Cluster).

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
