# Peer review of "One-form symmetries in $\mathcal{N} = 3$ $S$-folds"

_SciPost Physics, doi:SciPost Phys. 15, 132 (2023)_

## Round 1 · Referee Report · Max Hubner (Referee 1) · 2023-6-21

Strengths
1 - Concisely written, the authors provide a clear step by step outline of their analysis and subsequently execute it
2 - Checks, the authors consider various limiting cases of their results and match to existing literature
2 - Checks, the authors consider various limiting cases of their results and match to existing literature
Weaknesses
1 - The authors only discuss the line defects building the representation of the 1-form symmetry group and do not consider the symmetry operators acting on these
2 - The results have not been checked via alternative methods, for example, via a geometric analysis in a dual M-theory frame
2 - The results have not been checked via alternative methods, for example, via a geometric analysis in a dual M-theory frame
Report
This paper computes various charge lattices, case by case, for S-fold SCFTs (realized in IIB) at generic points of their Coulomb branch. Computations run via string junction considerations. The authors compute the lattice of genuine line defect operators modulo screening and determine maximal sub-lattices of mutually local lines thereof (so-called polarizations). From here they infer the 1-form symmetry groups for all S-fold SCFTs via Pontryagin duality. The topological symmetry operators acting on the line operators are not discussed. The authors further argue that for a fixed relative S-fold SCFT, with multiple distinct polarizations standard, construction for non-invertible duality defects apply.
I recommend this article for publications without further revision.
I recommend this article for publications without further revision.

---

## Round 1 · Referee Report · Anonymous (Referee 2) · 2023-6-27

Report
In this paper the authors obtain the one-form symmetries for N=3 SCFTs that can be constructed using S-folds. They also discuss the non-invertible symmetries that appear in these setups.
Since most of these theories do not admit a Lagrangian desciption, the authors adapt ideas in previous work using BPS quivers (2204.06495) to the current setting, where BPS quivers are not known but string junction methods are available.
The analysis is clear, the paper is well written, and the results and methods are very interesting, so I will be happy to recommend publication once the following minor point in the exposition is addressed:
Eq. (11) shows how the S-fold acts on strings connecting branes on neighbouring S-fold fundamental regions, but then in eq. (19) for instance the S-fold transformation acts on strings on arbitrarily separated fundamental regions. For the benefit of the readers, I would ask the authors to describe explicitly how these more general transformations are obtained.
Since most of these theories do not admit a Lagrangian desciption, the authors adapt ideas in previous work using BPS quivers (2204.06495) to the current setting, where BPS quivers are not known but string junction methods are available.
The analysis is clear, the paper is well written, and the results and methods are very interesting, so I will be happy to recommend publication once the following minor point in the exposition is addressed:
Eq. (11) shows how the S-fold acts on strings connecting branes on neighbouring S-fold fundamental regions, but then in eq. (19) for instance the S-fold transformation acts on strings on arbitrarily separated fundamental regions. For the benefit of the readers, I would ask the authors to describe explicitly how these more general transformations are obtained.

---

## Round 2 · Author Response

We thank the referees for their thoughtful reports.

---

## Round 2 · List of Changes

We addressed the referee's request in Report 2 by adding mathematical details and a reference in the paragraph preceding Eq. (19) on page 7.

---

## Editorial Decision

published